# Fluorescence-detection size-exclusion chromatography utilizing nanobody technology for expression screening of membrane proteins

Fei Jin[1,2], Cheng Shen[1,2], Yao Wang[1], Mengqi Wang[1], Minxuan Sun[1] & Motoyuki Hattori [1✉]

GFP fusion-based fluorescence-detection size-exclusion chromatography (FSEC) has been widely employed for membrane protein expression screening. However, fused GFP itself may occasionally affect the expression and/or stability of the targeted membrane protein, leading to both false-positive and false-negative results in expression screening. Furthermore, GFP fusion technology is not well suited for some membrane proteins, depending on their membrane topology. Here, we developed an FSEC assay utilizing nanobody (Nb) technology, named FSEC-Nb, in which targeted membrane proteins are fused to a small peptide tag and recombinantly expressed. The whole-cell extracts are solubilized, mixed with anti-peptide Nb fused to GFP for FSEC analysis. FSEC-Nb enables the evaluation of the expression, monodispersity and thermostability of membrane proteins without the need for purification but does not require direct GFP fusion to targeted proteins. Our results show FSEC-Nb as a powerful tool for expression screening of membrane proteins for structural and functional studies.

[1] State Key Laboratory of Genetic Engineering, Shanghai Key Laboratory of Bioactive Small Molecules, Collaborative Innovation Center of Genetics and Development, Department of Physiology and Biophysics, School of Life Sciences, Fudan University, Shanghai, China. [2]These authors contributed equally: Fei Jin, Cheng Shen. ✉email: hattorim@fudan.edu.cn

Biophysical and biochemical studies, especially the structural determination of membrane proteins, require stable and homogeneous sample preparation, the acquisition of which is often hindered by the poor expression and unstable nature of membrane proteins[1–3].

To overcome this issue, various methods have been developed[4–16]. In particular, following pioneering work on the application of GFP fusion techniques for membrane protein expression screening[12–14], GFP fusion-based fluorescence-detection size-exclusion chromatography (FSEC) has been widely utilized for rapid evaluation of the expression status and thermostability of membrane proteins from both eukaryotes and prokaryotes[15,16].

In GFP fusion-based FSEC, recombinantly expressed GFP-fused proteins can be detected by a fluorescence detector following size-exclusion chromatography. The resulting fluorescence chromatography profiles allow one to rapidly analyze the expression level, monodispersity, and stability of both unpurified and purified membrane proteins at a scale on the order of nanograms. GFP fusion-based FSEC, which is suited for the high-throughput screening of panels of orthologues, mutations and membrane proteins under different biochemical conditions, has been shown to be powerful for determining the structure of eukaryotic and prokaryotic membrane proteins by both cryo-EM and X-ray crystallography[17–25].

Nevertheless, several significant disadvantages of this method have also been recognized.

First, because GFP is a highly stable, soluble protein, its fusion sometimes causes false-positive hits in FSEC screening. In the case of such false-positive hits, GFP fusion proteins exhibit monodispersity by FSEC, but target membrane proteins may immediately aggregate or precipitate after the removal of GFP due to the instability of the target membrane protein alone[26]. Second, in addition to the issue of false positivity, GFP fusion also causes false negativity because it sometimes negatively affects the expression level[27,28]. Finally, depending on the membrane topology of the target membrane protein, the GFP fusion technique may be difficult to apply. For instance, GFP fusion technology is not well suited for application with bacterial membrane proteins where both N- and C-terminal ends are located at the periplasm because GFP tends to fail to fold properly at the periplasm and thus does not show fluorescence[29,30]. Likewise, eukaryotic Cys-loop receptors are also known to be unsuitable for either N- or C-terminal GFP fusion[31,32]. Thus, the insertion of GFP into the cytoplasmic loop is required for the application of GFP technology[31,32]. This finding indicates that the simple strategy of N- or C-terminal GFP fusion is not applicable to some eukaryotic membrane proteins; thus, the application of GFP fusion-based FSEC may require optimization of the position at which GFP is inserted.

To overcome such disadvantages, a GFP fusion-free FSEC method would be ideal, and a multivalent nitrilotriacetic acid (NTA) fluorescent probe called P3NTA was developed as a pioneering work of the GFP fusion-free FSEC method[9]. The P3NTA probe can bind the poly-histidine tag fused to a target membrane protein for detection by FSEC without the need for purification. However, since interactions of the P3NTA probe with poly-histidine-tagged proteins are relatively weak and nonspecific, endogenous proteins from host cells with multiple accessible histidine residues may seriously affect the detection of target proteins[33]. In particular, expression constructs of membrane proteins with high stability and monodispersity but relatively moderate expression are hard to identify from FSEC screening by P3NTA due to its relatively weak and nonspecific detection ability. However, such expression constructs would still be promising since structure determination by cryo-EM requires much

less purified protein than structure determination by X-ray crystallography[34].

To make further practical use of GFP fusion-free FSEC, we hypothesized that the application of other types of small peptide tags with high affinity and specificity would be ideal and that recent advances in nanobody (Nb) technologies for small peptides would meet such demands for GFP fusion-free FSEC.

Nb technology has been broadly utilized in laboratory research, clinical diagnosis and potential therapies[35]. Nbs, which are derived from the antigen-specific variable domain of the camelid heavy-chain antibody, have a molecular weight of 12–15 kDa and can be recombinantly expressed in bacteria with high yield.

Recently, the peptide tags ALFA and BC2 and the corresponding specific Nbs we refer to here as NbALFA and NbBC2, respectively, have been developed[33,36]. The ALFA tag (SRLEEELRRRLTE), designed de novo, forms a stable, hydrophilic and electroneutral α-helix in solution with an extremely high affinity of ~26 pM for NbALFA[33,37]. The de novo-designed sequence of ALFA is absent in common model organisms, which makes its recognition by NbALFA unique[33]. The BC2 tag (PDRKAAVSHWQQ), derived from residues 16–27 of β–catenin, is unstructured in solution and has a high affinity of ~1.4 nM for NbBC2[36].

Here, we developed a new type of FSEC utilizing Nb technology named FSEC-Nb. A membrane protein fused to the small peptide tag ALFA is recombinantly expressed in bacterial or eukaryotic cells. The whole-cell extracts are then solubilized and mixed with NbALFA Nb, which is specific for the ALFA tag, fused to mEGFP[38] for FSEC analysis (Fig. 1).

To validate the method, we applied FSEC-Nb to the expression of bacterial and eukaryotic membrane proteins and showed that FSEC-Nb can be applied to orthologue screening and a thermostability assay.

Notably, we applied FSEC-Nb to orthologues of the zinc-activated ion channel (ZAC) family, a member of the Cys-loop receptor superfamily, which are unsuitable for either N- or C-terminal GFP fusion, and identified a ZAC orthologue from Oryzias latipes (OlZAC). However, we were not able to detect the expression of OlZAC by P3NTA, a previously developed GFP fusion-free FSEC method. Consistent with the FSEC-Nb results, the negative staining EM and cryo-EM of the purified ZAC orthologue showed the monodispersity of the particles. Furthermore, we screened the expression of membrane proteins from SARS-CoV-2 by FSEC-Nb and identified two of them with a high level of expression and monodispersity, which could facilitate further structural and functional studies of SARS-CoV-2. Overall, our results showed FSEC-Nb to be a powerful tool for screening the expression of membrane proteins.

## Results

**Establishing the FSEC-Nb method**. To overcome the disadvantages of the conventional FSEC method, we designed FSEC-Nb, which utilizes short peptides as fusion tags and Nbs specific to these peptides fused to monomerized EGFP as a probe (Fig. 1).

We first applied our method to a prokaryotic orthologue of Zrt/Irt-like protein (ZIP) in the E. coli expression system. ZIPs function as metal transporters and are conserved from prokaryotes to eukaryotes, including humans[39]. Among the ZIP family, the structure of the bacterial ZIP protein from Bordetella bronchiseptica (BbZIP) was determined by crystallography[39]; we chose to utilize BbZIP to establish our FSEC-Nb system because both its N- and C-terminal ends are located at the periplasm[39], which is not well suited for application of the GFP fusion-based FSEC method in bacterial expression systems.

In our experiment, BbZIP was fused to the peptide tags ALFA and BC2 at its C-terminus and recombinantly expressed in E. coli.

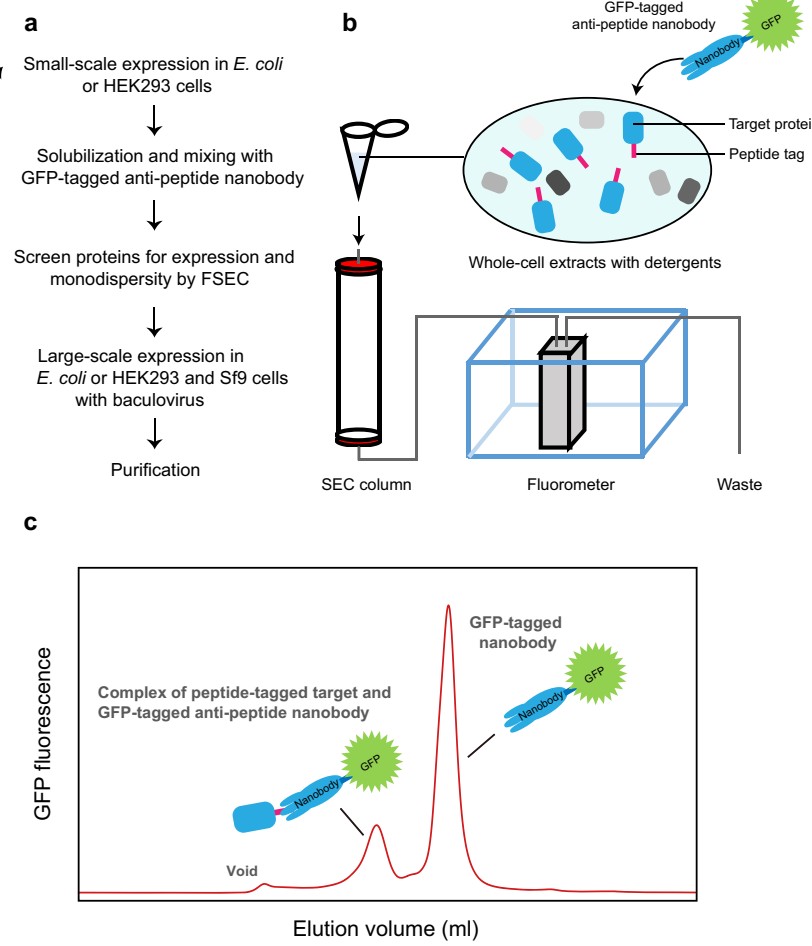

**Fig. 1 FSEC-Nb designation and verification. a** Flow chart of FSEC-Nb for membrane protein expression and purification. **b** Cartoon diagram of the FSEC system for FSEC-Nb. **c** Cartoon of a representative FSEC trace from FSEC-Nb.

The whole-cell extract was solubilized with detergents and mixed with mEGFP-fused Nbs specific for either the ALFA or BC2 tag (Fig. 2a, b). After removal of the pellet by ultracentrifugation, the sample was applied to an SEC column connected to a fluorescence detector (Fig. 1). When BbZIP was probed with mEGFP-tagged NbALFA, the FSEC plots presented peaks for both the mEGFP-tagged NbALFA in complex with the ALFA peptide-tagged BbZIP and free mEGFP-tagged NbALFA (Fig. 2a), but the corresponding complex peak was not observed when BbZIP was probed with mEGFP-tagged NbBC2 (Fig. 2b). These results showed that mEGFP-NbALFA specifically recognized the ALFA peptide-tagged BbZIP protein for the detection of BbZIP expression. The reason for the failure of mEGFP-tagged NbBC2 and the BC2 tag is unknown but may have been due to the difference between tags in terms of their affinities for their Nbs (ALFA: ~26 pM, BC2: ~1.4 nM)[33,36]. Furthermore, we tested the expression of the C-terminally mEGFP-tagged BbZIP by FSEC but did not detect its expression (Fig. 2c), consistent with the finding that the C-terminal end of BbZIP is located at the periplasm[39].

Furthermore, we performed FSEC-Nb experiments on ALFA-tagged BbZIP in *E. coli* cells by increasing the starting culture volume. As expected, the FSEC-Nb plots showed higher peaks with more shape features as the culture volume increased (Fig. 2d).

Overall, based on the results from BbZIP, we decided to employ the ALFA peptide tag and mEGFP-tagged NbALFA with our FSEC-Nb system for further experiments.

**Thermostability assay by FSEC-Nb**. We next applied the FSEC-Nb method to check membrane protein expression in mammalian cells and tested whether the FSEC-Nb system can be employed for thermostability assays of membrane proteins (Fig. 3a, b). We chose the human P2X3 (hP2X3) protein, a member of the P2X receptor superfamily with known structures[40–42].

ALFA-tagged hP2X3 was transiently expressed in HEK293 cells, which were solubilized for further FSEC-Nb experiments. The FSEC profiles of ALFA-tagged hP2X3 labelled with mEGFP-fused NbALFA showed peaks for both the mEGFP-fused NbALFA in complex with the ALFA-tagged hP2X3 and free mEGFP-fused NbALFA (Fig. 3c), showing that the FSEC-Nb technique can be applied in HEK293 cells. We also performed FSEC-Nb experiments of ALFA-tagged hP2X3 in HEK293 cells by increasing the culture volume (1 ml, 2 ml, and 4 ml), which resulted in higher peaks for hP2X3 as the culture volume increased (Fig. 3c). Intriguingly, as the peak became higher, the elution position of the peak shifted to a position corresponding to a lower molecular weight (Fig. 3c), probably because as the culture volume increased, the fractions of the P2X3 trimer, which does not have three full mEGFP-tagged NbALFAs bound per P2X3 trimer, increased, yielding a shift of the elution position to that corresponding to a lower molecular weight.

In the thermostability assay of hP2X3 by FSEC-Nb, solubilized samples were incubated at their respective temperatures for 10 min using a thermal cycler, and the precipitated materials were then removed by ultracentrifugation before labelling with

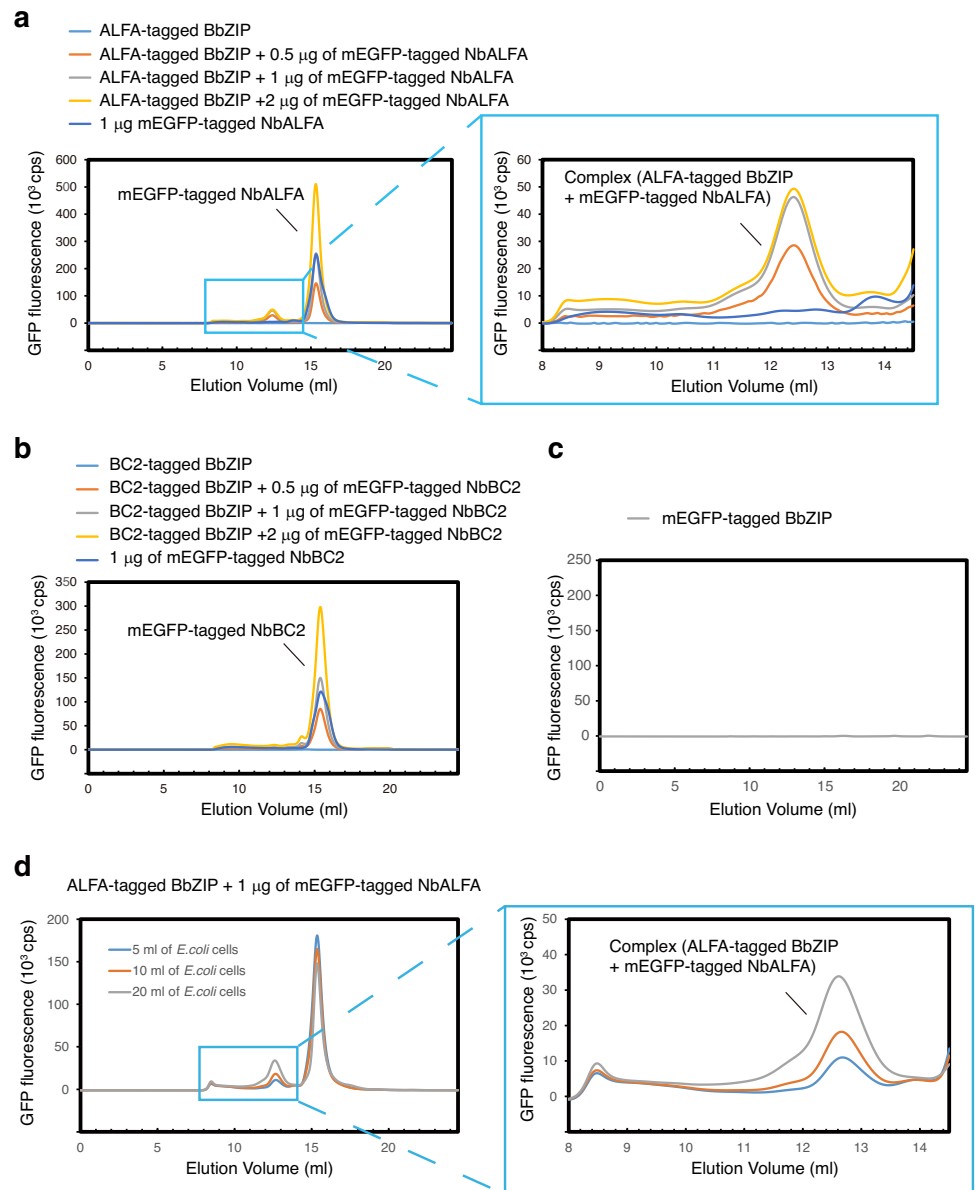

**Fig. 2 Establishing the FSEC-Nb method. a** FSEC traces of unpurified ALFA peptide-tagged BbZIP with mEGFP-tagged NbALFA, as detected by mEGFP fluorescence. A close-up view of the main peak profiles for the complex of ALFA-tagged BbZIP and mEGFP-tagged NbALFA is also shown. **b** FSEC traces of unpurified BC2 peptide-tagged BbZIP with mEGFP-tagged NbBC2, as detected by mEGFP fluorescence. **c** FSEC traces of C-terminally mEGFP-tagged BbZIP, as detected by mEGFP fluorescence. **d** FSEC-Nb traces of unpurified ALFA peptide-tagged BbZIP expressed in different volumes of *E. coli* cells with mEGFP-tagged NbALFA, as detected by mEGFP fluorescence.

mEGFP-tagged NbALFA (Fig. 3a, b). The FSEC-Nb profiles clearly showed a thermal shift of the main peaks from the samples incubated at temperatures near and above 55 °C (Fig. 3d), with an estimated $T_m$ of 56.6 °C.

We then tested the thermostabilizing effects of ATP on hP2X3 (Fig. 3e). ATP is an endogenous ligand of P2X receptors that typically increases the thermostability of P2X receptors[16]. Consistently, in the thermostability assay carried out by FSEC-Nb, ATP showed a clear stabilizing effect, increasing the estimated $T_m$ by 15 °C. These results showed that FSEC-Nb can be employed to assay the thermostability of membrane proteins without the need for purification steps.

**Expression screening of ZAC orthologues and SARS-CoV-2 membrane proteins.** As examples of the practical application of

FSEC-Nb, we then applied FSEC-Nb to screen ZAC family proteins and membrane proteins from SARS-CoV-2 (Figs. 4 and 5).

ZACs belong to the Cys-loop ligand-gated ion channel (LGIC) superfamily, which also includes nicotinic acetylcholine (nACh), 5-HT$_3$, GABA$_A$ and glycine receptors[43,44].

In addition to Zn$^{2+}$, the gating of ZACs, nonselective cation channels that are widely expressed in the human body, is activated by Cu$^{2+}$ and protons[44]. Since ZACs were the last members of the Cys-loop LGIC superfamily to be discovered[43–45], their function and structure are poorly characterized.

To facilitate structural and biophysical studies of ZAC proteins, we utilized FSEC-Nb to overcome the difficulty imposed by heterogeneous ZAC expression and purification. We chose to apply FSEC-Nb to ZACs because Cys-loop LGIC superfamily proteins were reported to be unsuitable for either N- or C-terminal GFP fusion[31,32].

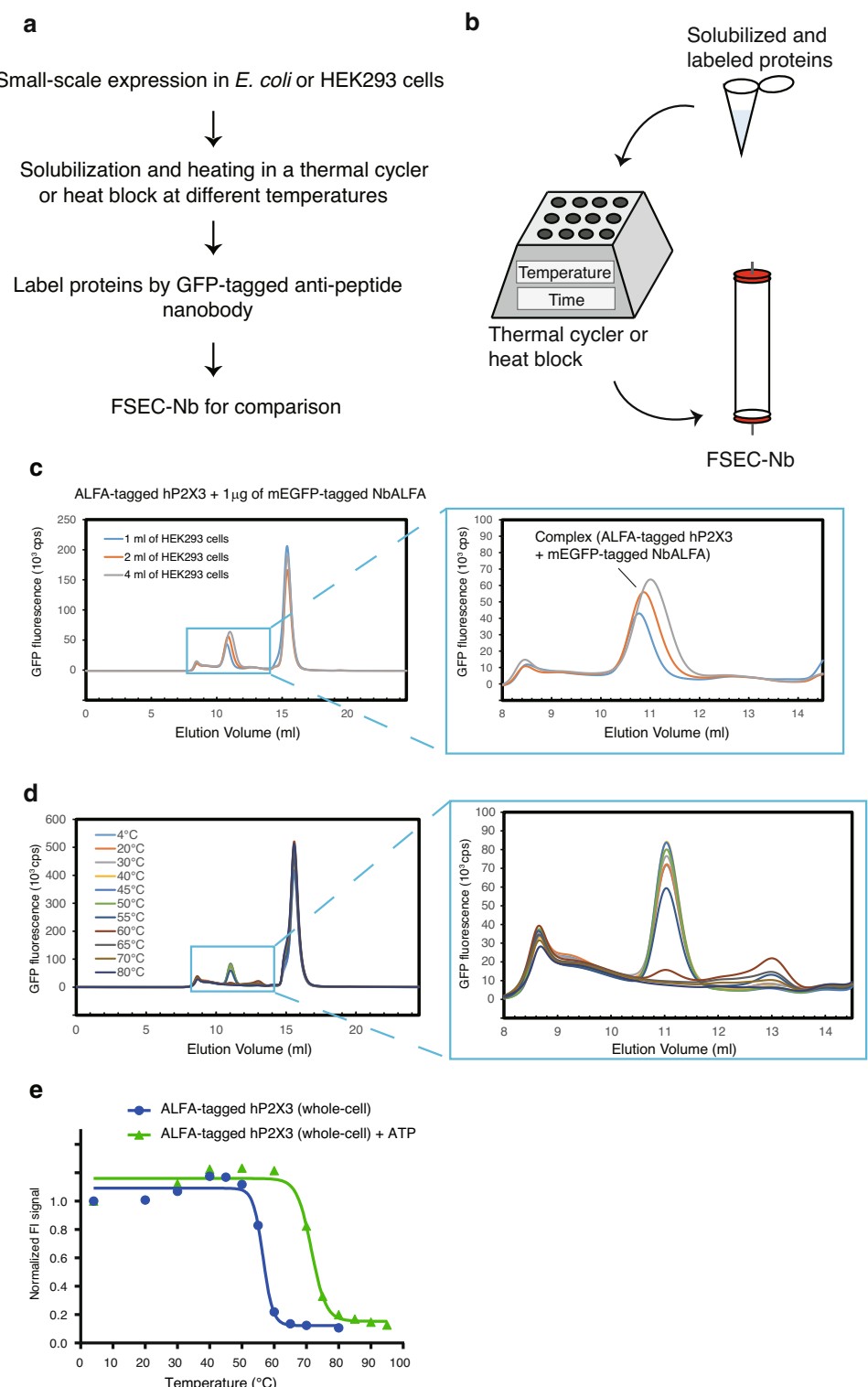

**Fig. 3 Thermostability assay by FSEC-Nb. a** Flow chart of the thermostability assay by FSEC-Nb. **b** Cartoon diagram of the thermostability assay by FSEC-Nb. **c** FSEC-Nb traces of unpurified ALFA peptide-tagged hP2X3 expressed in different volumes of HEK293 cells with mEGFP-tagged NbALFA, as detected by mEGFP fluorescence. A close-up view of the main peak profiles for the complex of ALFA-tagged hP2X3 and mEGFP-tagged NbALFA is also shown. **d** FSEC-Nb traces of unpurified ALFA-tagged hP2X3 preheated at the indicated temperatures. A close-up view of the main peak profiles for the complex of ALFA-tagged hP2X3 and mEGFP-tagged NbALFA is also shown. **e** Melting curves of hP2X3 in the presence and absence of ATP, as detected by FSEC-Nb. The fitted curves are shown as blue (apo) and green (with ATP) lines.

ZAC genes from *Homo sapiens* (HsZAC), *Danio rerio* (DrZAC), *Oryzias latipes* (OlZAC) and *Oreochromis niloticus* (OnZAC) were synthesized with ALFA and octa-histidine tags at the C-terminus and recombinantly expressed in HEK293 cells. The expressed ZAC orthologues were probed by mEGFP-tagged NbALFA for detection by the FSEC-Nb method. FSEC-Nb screening of ZAC orthologues showed that the profile for OlZAC exhibited a higher and sharper peak than the peaks for other ZAC orthologues (Fig. 4a).

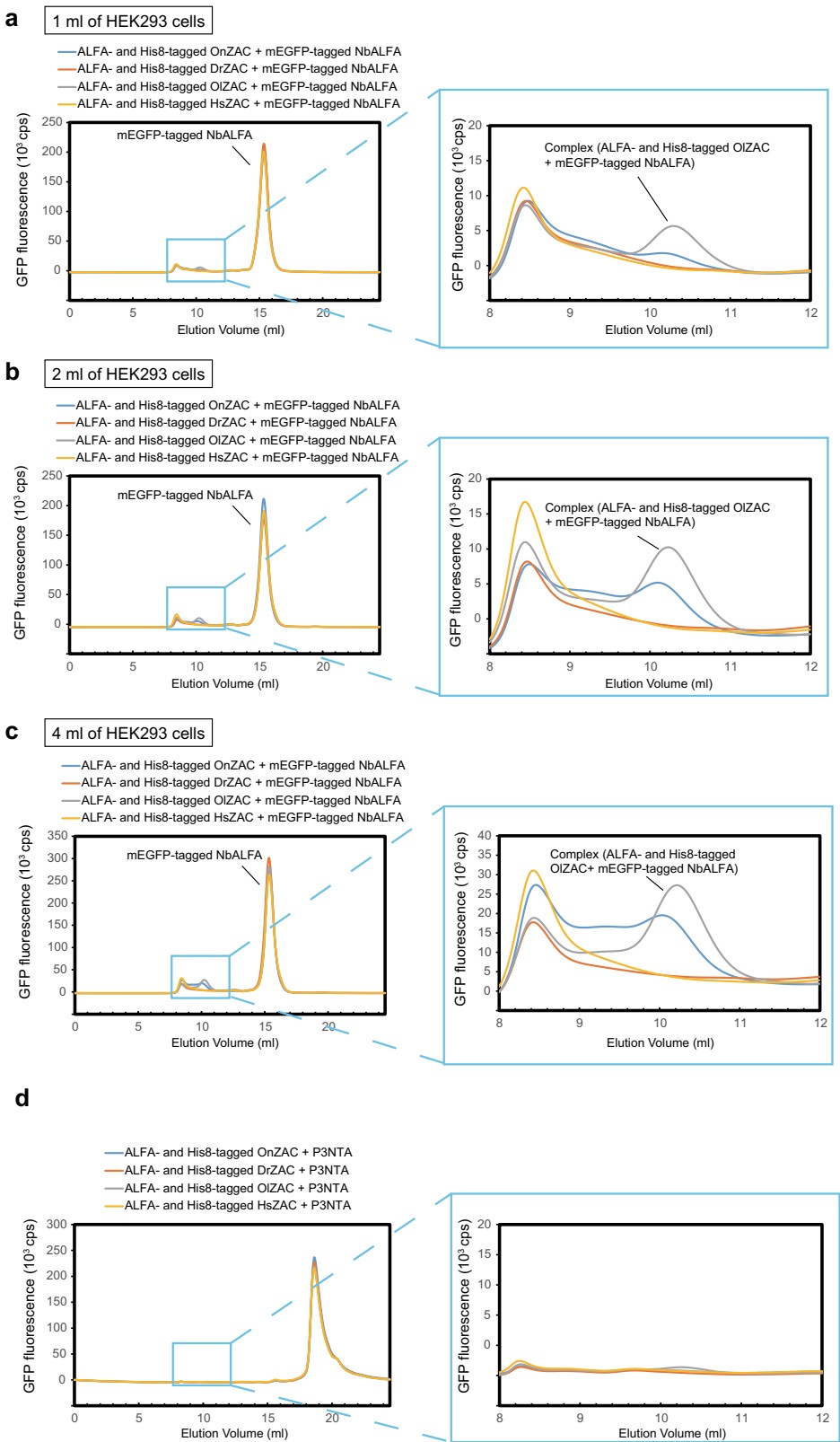

**Fig. 4 Expression screening of ZAC orthologues. a–c** FSEC-Nb traces of unpurified ALFA peptide and His8-tagged ZAC orthologues expressed in 1 ml (**a**), 2 ml (**b**) and 4 ml (**c**) scale cultures of HEK293 cells with mEGFP-tagged NbALFA, as detected by mEGFP fluorescence. A close-up view of the main peak profiles for the complex of ALFA-tagged ZAC and mEGFP-tagged NbALFA is also shown. The expression of ZAC orthologues from *Homo sapiens* (GI: 206725456), *Danio rerio* (528523664), *Oryzias latipes* (765127633), and *Oreochromis niloticus* (542233486) was screened by FSEC-Nb. **d** FSEC traces of unpurified ALFA peptide and His8-tagged ZAC orthologues with P3NTA, as detected by fluorescein fluorescence.

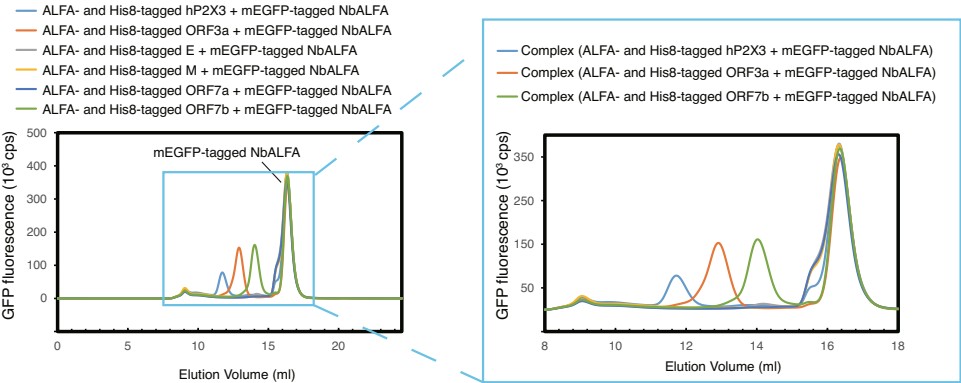

**Fig. 5 Expression screening of membrane proteins from SARS-CoV-2.** FSEC-Nb traces of unpurified ALFA peptide and His8-tagged membrane proteins from SARS-CoV-2 with mEGFP-tagged NbALFA, as detected by mEGFP fluorescence. A close-up view of the main peak profiles is also shown. The expression of ORF3a (UniProt ID: P0DTC3), E (P0DTC4), M (P0DTC5), ORF7a (P0DTC7), and ORF7b (P0DTD8) was screened by FSEC-Nb.

We also performed FSEC-Nb experiments to screen ALFA-tagged ZAC homologs by increasing the starting culture volume (1 ml, 2 ml, and 4 ml), which resulted not only in higher peaks but also in clearer peak shapes from ZAC homologues (Fig. 4a–c). With 2 ml and 4 ml of the starting culture volume, OlZAC exhibited the high monodispersity of the main peak in FSEC-Nb (Fig. 4b, c). In contrast, the FSEC plots of OnZAC showed a much broader peak with the left shoulder in FSEC-Nb (Fig. 4b, c). Consistently, we could not purify OnZAC as it aggregated during purification. Thus, FSEC-Nb can be used to distinguish protein samples with good behavior from poorly behaved samples based on the monodispersity of peak shapes in FSEC-Nb.

Furthermore, we could not detect the expression of OlZAC by P3NTA-based FSEC, the previously developed GFP fusion-free FSEC method (Fig. 4d), showing the improved sensitivity of FSEC-Nb over P3NTA-based FSEC.

SARS-CoV-2 is a pathogen that causes coronavirus disease 2019 (COVID-19)[46–48]. Using FSEC-Nb, we screened the expression of a series of membrane proteins from SARS-CoV-2 (Fig. 5). We identified ORF3a and ORF7b with high monodispersity and high expression levels, comparable to those of hP2X3 (Fig. 5). ORF3a is an ion channel and potential target for COVID-19 therapy[49]. A mutation in ORF7b reportedly showed higher replicative fitness[50]. Consistent with the sharp peak from FSEC-Nb, the cryo-EM structure of ORF3a was recently reported on bioRxiv[49]. Overall, our FSEC-Nb screening results may facilitate structural and functional studies of SARS-CoV-2.

**Detergent screening of OlZAC.** Purification of membrane proteins requires detergents to extract the proteins from the biological membrane. The type of detergent used often affects the monodispersity and stability of a membrane protein in purification; thus, detergent screening is beneficial for establishing purification protocols for membrane proteins. Furthermore, the addition of lipids and lipid-like compounds such as cholesteryl hemisuccinate (CHS), which was shown to be useful for the purification and crystallization of various GPCRs[51–53], could also affect the stability of membrane proteins[16]. In our assay of OlZAC thermostability by FSEC-Nb, we tested multiple types of detergents for OlZAC; among these detergents were n-dodecyl-b-D-maltoside (DDM); DDM additive with CHS at a ratio of 5:1 (w:w), referred to as DDM-CHS; lauryl maltose neopentyl glycol (LMNG); and glycodiosgenin (GDN). The FSEC-Nb profiles of OlZAC solubilized with DDM showed a thermal shift of the main peaks from the samples incubated at temperatures near and above 60 °C (Fig. 6a), with estimates of the Tm of 60.6 °C (Fig. 6b). Unpurified ALFA-tagged OlZAC samples solubilized with the respective detergents were heat-treated at 60 °C

for 10 min, and FSEC-Nb was applied to both heated and unheated samples for comparison (Fig. 6c). Compared to DDM, LMNG conferred better thermostability to OlZAC, whereas DDM-CHS solubilized OlZAC similarly as with DDM (Fig. 6c). The performance of GDN was similar to the performance of LMNG (Fig. 6c). Considering the results of detergent screening with OlZAC by FSEC-Nb, we decided to employ either DDM or DDM-CHS for protein extraction from the membrane and either LMNG or GDN for the subsequent purification steps.

**Large-scale culture and purification of OlZAC.** For large-scale culture in HEK293S cells, we then generated bacmid DNA for OlZAC, which was used to transfect Sf9 insect cells to prepare BacMam virus. To optimize the expression conditions, using FSEC-Nb, we performed small-scale expression screening in HEK293S cells by testing different amounts of P2 virus, incubation times, and cell culture temperatures at 16 h after the addition of P2 virus (Fig. 7a, b) and decided to choose one set of conditions for large-scale culture (1% volume P2 virus addition, 88 h of culture at 37 °C after the addition of virus).

OlZAC was purified as described in the Methods section. Briefly, the membrane collected from cell lysates was solubilized in DDM-CHS, and the detergent was then exchanged to LMNG in affinity chromatography steps. During size-exclusion chromatography in SEC buffer containing LMNG, the UV absorbance plot showed a symmetrical peak for OlZAC and a prior void peak (Fig. 7c). A total of 2.4 litres of HEK293 cell culture yielded ~0.5 mg of purified OlZAC protein. Trp-based FSEC verified the monodispersity of the pooled fractions constituting the main SEC peaks (Fig. 7d), and the purity of the pooled fractions was validated by SDS-PAGE (Fig. 7e).

**Negative staining EM and cryo-EM of OlZAC.** To evaluate the sample quality of OlZAC, which was identified by FSEC-Nb, we performed negative staining EM and preliminary cryo-EM of OlZAC (Fig. 8).

Pioneering structural studies of other eukaryotic pLGIC members by crystallography and cryo-EM have elucidated their fundamental architecture: a pentamer comprised of an extracellular component for ligand gating, a transmembrane component for ion permeation and an intracellular component[32,54–60]. ZACs possess low amino acid sequence identity with other pLGIC members, with the closest matches exhibiting ~20% identity with ZACs[43,44]. Accordingly, little is still known about the ZAC structure.

The OlZAC purified under apo conditions was reconstituted into amphipol by mixing with amphipols at a mass ratio of 1:20, and the detergent was removed by Bio-Beads. We tested the

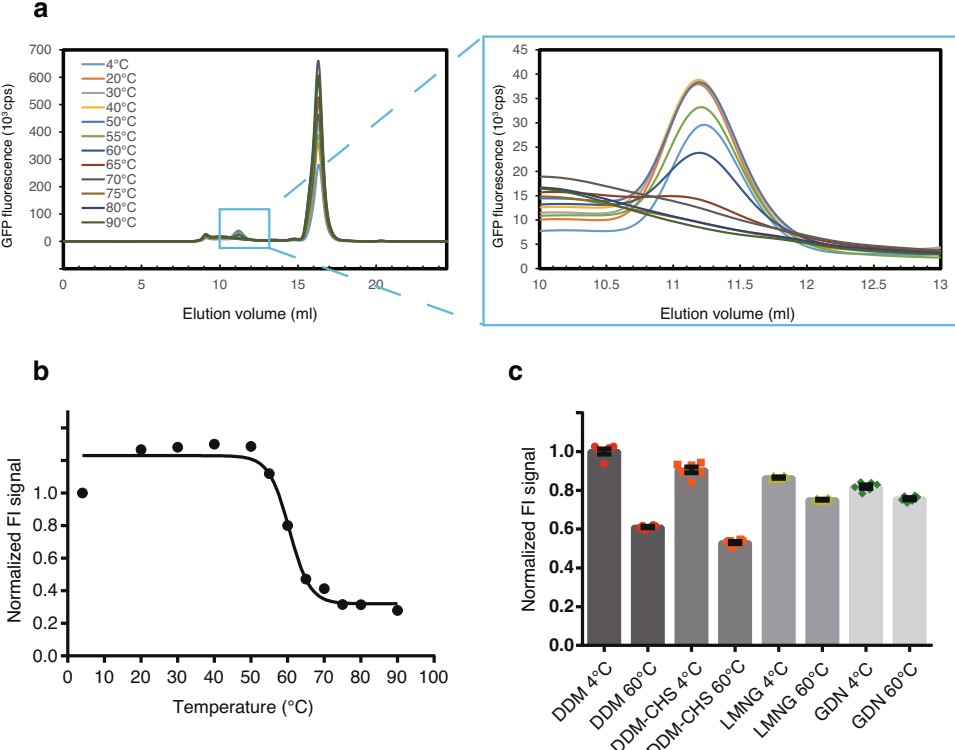

**Fig. 6 Detergent screening for OlZAC purification. a** FSEC-Nb traces of unpurified ALFA-tagged OlZAC preheated at the indicated temperatures. A close-up view of the main peak profiles is also shown. **b** Melting curves of OlZAC, as detected by FSEC-Nb. The fitted curve is shown as a black line. **c** Normalized peak heights of ALFA-tagged OlZAC preheated at 60 °C for 10 min solubilized with the indicated detergents. The peak heights were normalized to the peak from the sample solubilized with DDM at 4 °C. Error bars represent the standard error of the mean (N = 6).

reconstitution of NAPol on a small scale by Trp-FSEC, which resulted in a high and symmetrical peak for the amphipol-reconstituted OlZAC (Supplementary Fig. 1a). NAPol is a nonionic amphipol that is soluble across a wide pH range and compatible with multivalent cations[61,62]; thus, we chose NAPol for ZACs since both pH and the presence of divalent cations are relevant to the functional status of ZACs. On a large scale, we further reconstituted OlZAC into NAPol and separated the amphipol-reconstituted OlZAC by SEC (Supplementary Fig. 1b).

The amphipol-reconstituted OlZAC was then stained with uranyl acetate and observed under an electron microscope. The images taken by the EM-CCD camera showed monodispersed OlZAC particles (Fig. 8a). Because of the high contrast after negative staining, the particles were easily recognized from the images (Fig. 8b). The particles extracted from over one hundred images were classified into several 2D classes, which validated the stoichiometry and constitution of ZACs (Fig. 8c–e). Similar to other pLGIC members, ZACs form a pentamer (Fig. 8e) and are composed of extracellular, transmembrane and intracellular components (Fig. 8d).

We then performed preliminary cryo-EM single-particle analysis of OlZAC with a K3 direct detection camera (Fig. 8f, g), which also showed monodispersed particles.

These results showed the sample quality of OlZAC identified by FSEC-Nb, which would be suitable for structural studies.

## Discussion
In this work, we developed a new type of FSEC assay, named FSEC-Nb, utilizing the ALFA peptide tag and anti-ALFA peptide Nb NbALFA. In FSEC-Nb, targeted membrane proteins are tagged by the peptide tag ALFA and recombinantly expressed in

either prokaryotic or eukaryotic cells before being probed by mEGFP-tagged NbALFA for FSEC analysis (Fig. 1). We first tested two peptide tags, ALFA and BC2, and found that the peptide tag ALFA was more suitable for the detection of BbZIP by FSEC-Nb in a bacterial expression system (Fig. 2). Although we did not test other peptide tags and their nanobodies for FSEC-Nb, it is possible that other peptide tags can be successfully applied for FSEC-Nb. We then applied the FSEC-Nb method for a thermo-stability assay (Fig. 3). As a demonstration of the practical application of FSEC-Nb, we then applied FSEC-Nb for the screening of orthologues of ZAC, a member of the Cys-loop LGIC superfamily without a known 3D structure, as well as membrane proteins from SARS-CoV-2 (Figs. 4, 5). We then further screened different types of detergents for the purification of OlZAC (Fig. 6). Finally, using purified OlZAC (Fig. 7), we performed negative staining EM and preliminary cryo-EM, which showed the monodispersity of the purified OlZAC sample (Fig. 8).

FSEC-Nb confers the advantage of conventional GFP fusion-based FSEC but avoids the following disadvantages of GFP fusion-based FSEC.

First, GFP fusion-based FSEC is not well suited for some membrane proteins, depending on their membrane topology[29–32]. Notably, the membrane topology prediction from 29 organisms by the TransMembrane Hidden Markov Model[63] showed that ~20% of multispanning membrane proteins possess both their N- and C-terminal ends at the extracellular or periplasmic sides. Furthermore, even when applicable, GFP fusion may occasionally affect the expression and/or stability of targeted membrane proteins, poten-tially leading to both false-positive and false-negative results[26–28]. In our recent worst case, we screened over 60 homologues of MgtC, a virulence factor in *Salmonella enterica*[64], by GFP fusion-based FSEC with C-terminal mGFPuv-tagged expression constructs

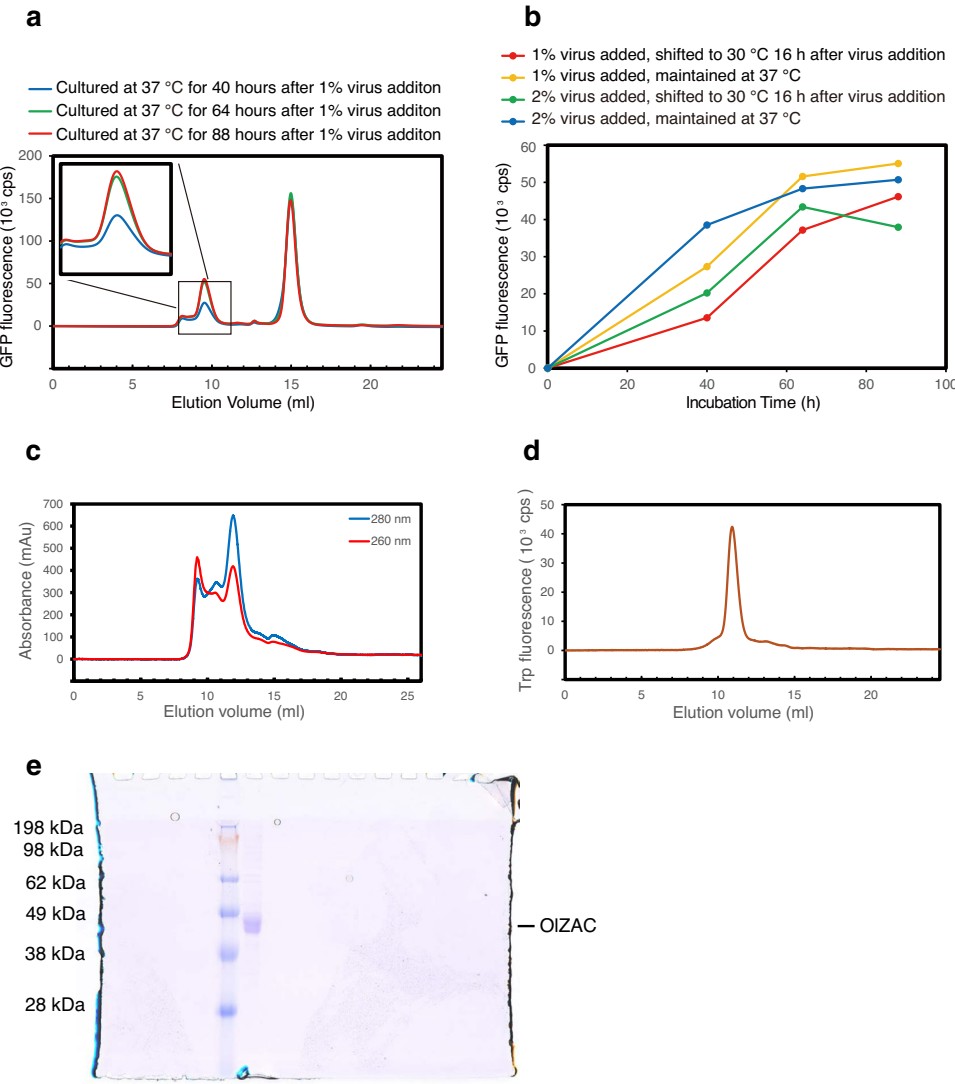

**Fig. 7 Large-scale expression and purification of OlZAC. a** FSEC profiles of OlZAC, as detected by FSEC-Nb for the optimization of cell culture conditions. **b** Time course curves of the main peak heights, as detected by FSEC-Nb. HEK293S cells were infected with P2 BacMam virus for OlZAC expression at a 1% or 2% volume. At 16 h after virus addition, cell culture temperatures were maintained at 37 °C or shifted to 30 °C. **c** Size-exclusion chromatography of OlZAC, as detected by UV absorbance. **d** FSEC trace of purified OlZAC, as detected by Trp fluorescence. **e** SDS-PAGE of the purified OlZAC after SEC.

(Supplementary Table 1) because the C-terminal end of MgtC is located in the cytoplasm. We identified only two with high monodispersity and expression levels (Supplementary Fig. 2a, b). However, both proteins aggregated and precipitated after the removal of the GFP tag. Consistently, we did not detect the expression of these two proteins at the corresponding positions by FSEC-Nb (Supplementary Fig. 2c, d). In addition, compared to the P3-NTA method, a previously developed GFP fusion-free FSEC method utilizing poly-histidine tag, FSEC-Nb, showed better performance in the screening of ZAC protein orthologues (Fig. 4). Overall, FSEC-Nb would be useful for expression screening of both types of membrane proteins to which the conventional GFP fusion-based FSEC is applicable and is not applicable.

On the other hand, representing a disadvantage of our FSEC-Nb assay over the conventional GFP fusion FSEC method, purified GFP-fused NbALFA, which acts as a probe, needs to be prepared in each laboratory that wishes to use this method. However, the purification of mEGFP-fused NbALFA would be easy for most biochemistry and structural biology laboratories, as its *E. coli* expression level is quite high (more than 10 mg of purified protein from 1 litre of *E. coli* culture), and 1 mg of mEGFP-fused NbALFA

is enough for 1000 FSEC-Nb experiments and would thus last for some years with conventional laboratory usage. Furthermore, to improve access for FSEC-Nb, we deposited the expression vectors for mEGFP- and mCherry-fused NbALFAs as well as template vectors with the ALFA tag for expression in *E. coli* (pETNb-nALFA and pETNb-cALFA) and insect (pFBNb-cALFA) and mammalian (pBMNb-cALFA) cells to the Addgene plasmid repository (Fig. 9). We also deposited the BbZIP gene in pETNb-cALFA and the hP2X3 gene in pBMNb-cALFA as positive controls for FSEC-Nb. Thus, FSEC-Nb can easily be introduced to most biochemistry and structural biology laboratories, particularly to laboratories with experience with the conventional GFP fusion-based FSEC method, which has already been widely used. Notably, all of these vectors can be used for not only a small-scale expression check by FSEC-Nb but also large-scale protein expression.

Overall, FSEC-Nb can be used for expression screening and thermostability assays on a small scale with high sensitivity and specificity without the need for GFP fusion to target proteins. Such advantages of FSEC-Nb will enable us to explore further opportunities to prepare target proteins for structure determination as well as other biophysical and pharmacological studies.

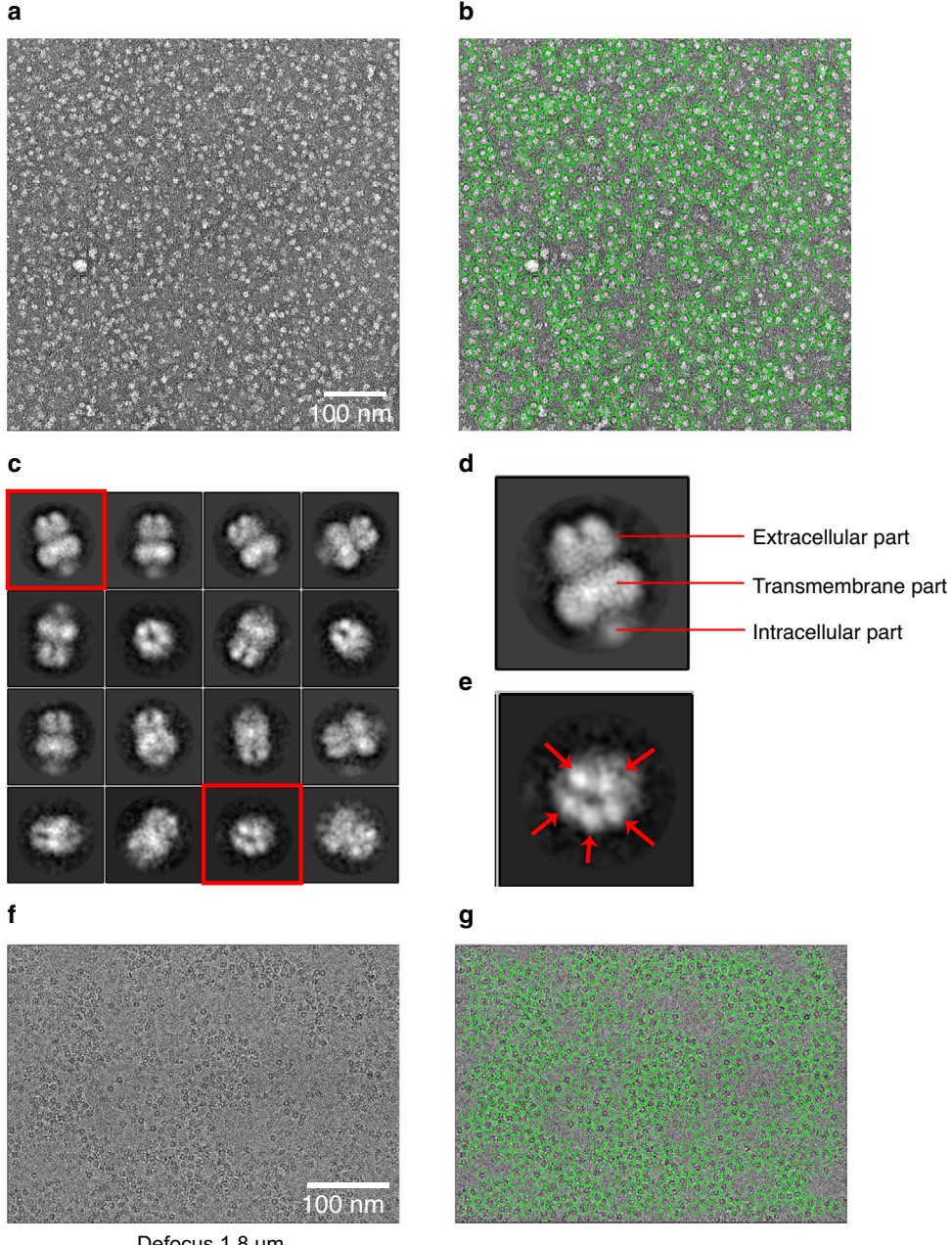

**Fig. 8 Negative staining EM and cryo-EM of OIZAC. a, b** Negative staining EM images of OIZAC particles. **c, d, e** Selected 2D class averages, as calculated using RELION. **f, g** Cryo-EM image at 1.8 μm defocused with OIZAC particles.

## Methods

**Purification of mEGFP-tagged Nbs**. With an interval of GSGSGS, the NbALFA sequence was fused in frame with an N-terminal His$_8$-mEGFP affinity tag and subcloned into the pET28b vector. The protein was overexpressed in *E. coli* Rosetta (DE3) cells in LB medium containing 30 μg/ml kanamycin at 37 °C by induction at an OD$_{600}$ of ~0.5 with 0.5 mM isopropyl D-thiogalactoside for 16 h at 18 °C. The *E. coli* cells were subsequently harvested by centrifugation (6000 × *g*, 15 min) and resuspended in buffer A (50 mM Tris-HCl (pH 8.0), 150 mM NaCl) supplemented with 0.5 mM phenylmethylsulfonyl fluoride (PMSF). All purification steps were performed at 4 °C. The *E. coli* cells were then disrupted with a microfluidizer, and debris was removed by centrifugation (70,000 × *g*, 60 min). The supernatant was loaded onto equilibrated Ni-NTA resin pre-equilibrated with buffer A and mixed for 1 h. The column was then washed with buffer A containing 30 mM imidazole, and the protein was eluted with buffer A containing 300 mM imidazole. The imidazole was removed by dialysis in buffer B (20 mM HEPES (pH 7.0), 150 mM NaCl) overnight. Finally, the purified mEGFP-tagged NbALFA was concentrated to 1 mg/ml using an Amicon Ultra 30 K filter (Merck Millipore) and stored at −80 °C before use. mEGFP-tagged NbBC was similarly expressed and purified.

**FSEC-Nb in the *E. coli* expression system**. In the *E. coli* expression system, BbZIP tagged with either the ALFA or BC2 peptide at its C-terminus was synthesized and subcloned into the pET28b vector and overexpressed with a protocol similar to the protocol for the expression of mEGFP-tagged NbALFA described above. The *E. coli* cell pellets from 5 ml, 10 ml or 20 ml of LB culture were suspended in 400 μl of buffer A and sonicated, and the cell debris was removed by centrifugation (20,000 × *g*, 10 min). The lysates were solubilized by mixing 500 μl of buffer A containing 2% (w:v) DDM and 0.5 mM PMSF for 1 h and were ultracentrifuged (200,000 × *g*, 20 min). Unless noted, 1 μg of mEGFP-tagged NbALFA was added to the supernatant and incubated for 30 min. Considering the reported affinity values of NbALFA binding to an ALFA tag ($k_{on}$ (M$^{-1}$s$^{-1}$): 3.6 (±0.1) × 10$^5$, $k_{off}$ (s$^{-1}$): 9.4 (±0.2) × 10$^{-6}$, K$_D$ 26 (±1) pM)[33], 30 min of incubation would be enough for saturation of the binding. After centrifugation (20,000 × *g*, 10 min), 50 μl of the sample was applied to a Superdex 200 Increase 10/300 GL column (GE Healthcare) equilibrated with buffer A containing 0.05% (w:v) DDM for the FSEC assay. GFP fusion-based FSEC was performed similarly but without the labelling step with mEGFP-tagged NbALFA.

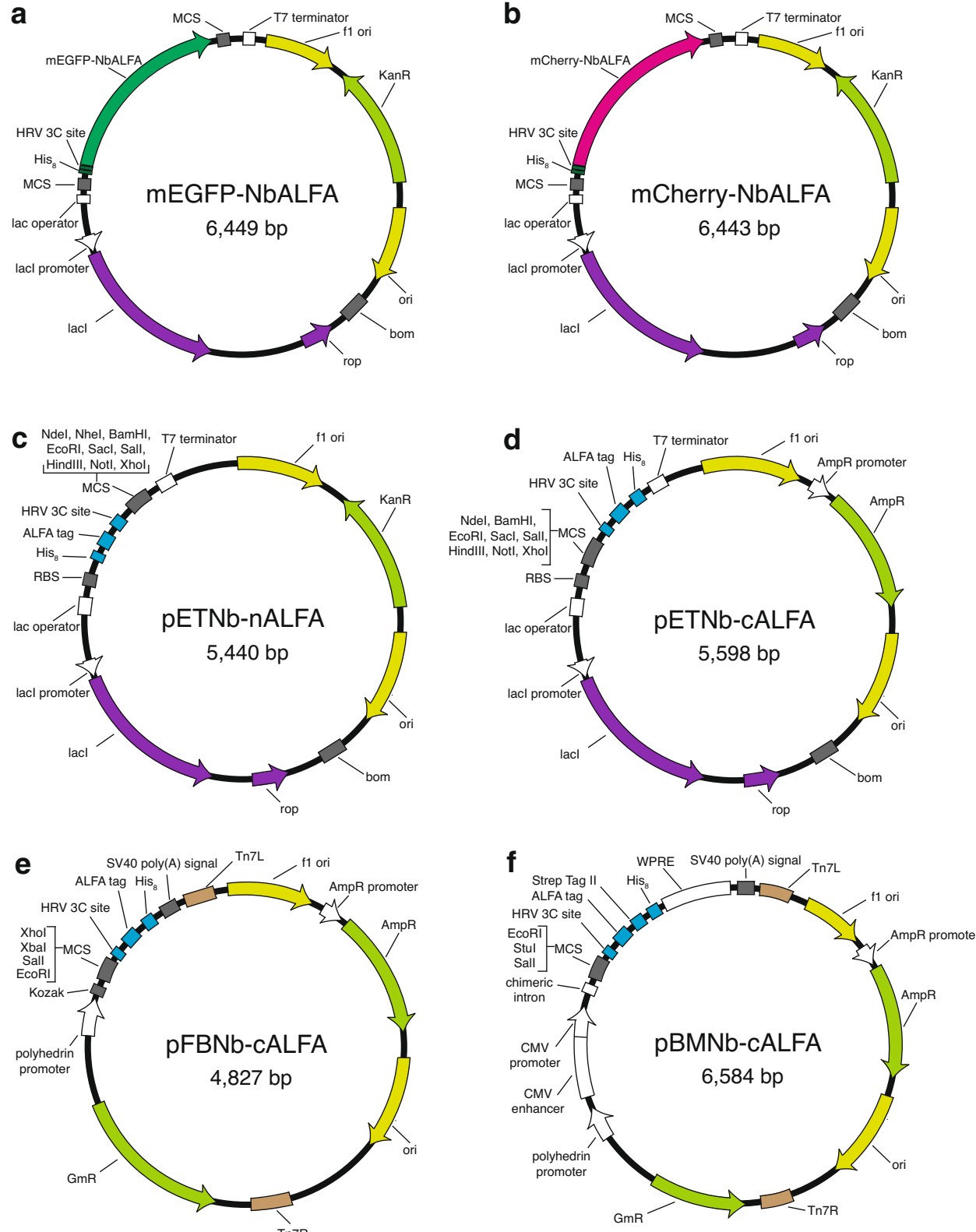

**Fig. 9 Vector maps for FSEC-Nb. a, b** Maps of the expression vectors for mEGFP-tagged (**e**) and mCherry-tagged (**f**) NbALFA. **c, d, e, f** Maps of the expression vectors for FSEC-Nb in *E. coli* (**c, d**), insect cells (**e**), and mammalian cells (**f**).

In the FSEC assay, fluorescence was detected using the RF-20Axs fluorescence detector for HPLC (Shimadzu, Japan) (for mEGFP, excitation: 480 nm, emission: 512 nm) (for mGFPuv, excitation: 395 nm, emission: 507 nm). FSEC-Nb experiments with mEGFP-tagged NbBC2 were performed with a similar protocol. FSEC-Nb experiments with MgtC were performed similarly.

**FSEC-Nb in the HEK293 expression system**. hP2X3, ZAC orthologues and membrane proteins from SARS-CoV-2 containing ALFA and His8 tags at their C-termini were synthesized and subcloned into a derivative of the Bac-to-Bac system vector with the CMV promoter and WPRE motif. Using 3 μl of Lipofectamine 2000 (Thermo Fisher Scientific, US), 1 μg of each plasmid per 1 ml cell culture was

transfected into 1 ml, 2 ml or 4 ml of HEK293S cells in adherent culture at a density of 0.5 million cells/ml in DMEM supplemented with 10% FBS. Cells were incubated in a $CO_2$ incubator (37 °C, 5% $CO_2$) for 48 h after transfection and solubilized with 200 μl of buffer A containing 2% (w:v) DDM supplemented with 0.5 mM PMSF, 5.2 μg/ml aprotinin, 2 μg/ml leupeptin, and 1.4 μg/ml pepstatin A (all from Sigma-Aldrich) for 1 h. After ultracentrifugation (200,000 × $g$, 20 min), 1 μg of mEGFP-tagged NbALFA or 1 μl of 50 μM P3NTA was added to and mixed into 100 μl of the supernatant for 30 min. Then, after centrifugation (20,000 × $g$, 10 min), 50 μl of the sample was applied to a Superdex 200 Increase 10/300 GL column (GE Healthcare) equilibrated with buffer A containing 0.05% (w:v) DDM for the FSEC assay. In the FSEC assay, fluorescence was detected as described above. The P3NTA peptide was prepared and used for FSEC as previously described (excitation: 480 nm, emission: 520 nm)[9].

**Thermostability assay by FSEC-Nb**. ALFA peptide-tagged hP2X3 was expressed in HEK293 cells and solubilized as described above. Cells from 4 ml of culture were resuspended in 1.2 ml of buffer A (50 mM TRIS (pH 8.0), 150 mM NaCl) containing 2% DDM by the addition of either ATP at a final concentration of 1 mM (ATP-bound conditions) or 0.6 units of apyrase (Sigma-Aldrich, USA) to remove endogenous ATP (apo conditions), rotated at 4 °C for 1 h, and then ultracentrifuged (200,000 $g$, 10 min). One hundred microlitres of the supernatant was dispensed into 1.5 ml Eppendorf tubes and incubated at the respective temperature for 10 min using either a thermal cycler or heat block bath. After ultracentrifugation (200,000 $g$, 10 min), the supernatant was mixed with 1 μg of mEGFP-tagged NbALFA and then centrifuged (20,000 $g$, 10 min). Then, 50 μl of the supernatant was applied to a Superdex 200 Increase 10/300 GL column (GE Healthcare) equilibrated with buffer A containing 0.05% (w:v) DDM for the FSEC assay. We estimated the melting curves based on the peak heights and determined the melting temperatures by fitting the melting curves to a sigmoidal dose-response equation because the melting curves based on the peak heights were known to be consistent with the melting curves based on the peak area estimated by Gaussian fitting[16].

**Detergent screening by FSEC-Nb**. HEK293S cells expressing ALFA-tagged OlZAC were prepared as described above. The collected cells were solubilized in buffer A containing different types of detergents: 2% (w:v) DDM, 2% (w:v) DDM-CHS, 1% (w:v) LMNG, and 1% (w:v) GDN. FSEC-Nb and thermostability assays by FSEC-Nb were conducted as described above.

**Expression and purification of OlZAC**. OlZAC tagged with ALFA and His$_8$ was expressed in HEK293S GnTI⁻ cells using a baculovirus-mediated gene transduction system in mammalian cells[65].

Small-scale expression screening to determine large-scale culture conditions was performed by FSEC-Nb (Fig. 7a, b). FSEC-Nb was carried out with the protocol described above. An 800 ml culture of HEK293S GnTI⁻ cells was grown to a density of $2.5 \times 10^6$ ml$^{-1}$ and infected with 8 ml of P2 BacMam virus. After 16 h of culture at 37 °C, 10 mM sodium butyrate was added, and the temperature was maintained at 37 °C for another 72 h of culture. Then, the cells were harvested and washed with buffer A. All purification steps were performed at 4 °C. Cells were lysed by sonication with protease inhibitors (1 mM PMSF, 5.2 μg/ml aprotinin, 2 μg/ml leupeptin, and 1.4 μg/ml pepstatin A, all from Sigma-Aldrich, USA). Membrane fractions were collected by ultracentrifugation (200,000 × $g$, 60 min). The membrane was solubilized in buffer A containing 2% (w:v) DDM-CHS and supplemented with protease inhibitors (1 mM PMSF, 5.2 μg/ml aprotinin, 2 μg/ml leupeptin, and 1.4 μg/ml pepstatin A, all from Sigma-Aldrich, USA) for 2 h. The debris was removed by ultracentrifugation (200,000 × $g$, 60 min). The supernatant was loaded onto equilibrated TALON resin (Takara, JAPAN) and then washed with buffer A containing 0.01% (w:v) LMNG and 10 mM imidazole. Protein was eluted with buffer A containing 300 mM imidazole. The eluted protein was loaded on a Superdex 200 10/300 GL column and subjected to SEC in buffer B containing 0.01% (w:v) LMNG. The main peak fractions were pooled and concentrated to ~1 mg/ml using an Amicon Ultra 100K filter (Merck Millipore).

**Amphipol reconstitution**. All steps were performed at 4 °C. On a small scale, 10 μg of OlZAC (10 μl) was mixed with 200 μg of NAPol (Anatrace, dissolved in 2 μl of buffer B) and incubated for 16 h. The detergent was removed by incubation with Bio-Beads SM-2 (Bio-Rad, USA) for 4 h, after which the beads were removed over a disposable Poly-Prep column. Twenty microlitres of the eluent was diluted to 200 μl, and 50 μl of the sample was applied to a Superdex 200 10/300 GL column equilibrated with buffer A for Trp-based FSEC (excitation: 280 nm, emission: 325 nm). At a large scale, 500 μg of OlZAC (500 μl) was mixed with 10 mg of NAPol (Anatrace, dissolved in 100 μl of buffer B) and incubated for 16 h. The detergent was removed with Bio-Beads SM-2 (Bio-Rad, USA) for 4 h, and the beads were subsequently removed over a disposable Poly-Prep column. The eluent was applied to a Superdex 200 10/300 GL column equilibrated with buffer B, and the main fractions consisting of the amphipol-reconstituted OlZAC were pooled and concentrated to ~3 mg/ml using an Amicon Ultra 100K filter for electron microscopic analysis.

**Negative staining and electron microscopy**. Gilder 400 square mesh grids (AG400) were glow-discharged in a PELCO easiGlow apparatus at a current of 25 mA for 30 s. Five microlitres of protein solution (~20 μg/mL) was dropped onto the grid and allowed to remain on the grid for 1 min. The residual protein solution was blotted from the grid edge with a piece of filter paper. The grid was covered with 2% uranyl acetate, blotted immediately, covered again with 2% uranyl acetate for 30 s and blotted again. After drying, the grid was observed under a Talos L120C microscope at 120 kV. In total, 133 micrographs were taken with a Ceta CCD camera at a nominal magnification of ×92,000 at a pixel size of 1.55 Å. The micrographs were processed in RELION 3.0 for particle picking, extraction and 2D classification[66].

**Cryo-EM data acquisition**. A total of 2.5 μl of OlZAC in NAPol was applied to a glow-discharged holey carbon film grid (QUANTIFOIL, R1.2/1.3, 100 Holey Carbon Films, Au 300 mesh) blotted with a Vitrobot (FEI) system using a 3.0 s blotting time with 100% humidity at 9 °C and plunge-frozen in liquid ethane. Cryo-EM images were collected on a Titan Krios (FEI) electron microscope operated at an acceleration voltage of 300 kV. The specimen stage temperature was maintained at 80 K. Images were recorded with a K3 Summit direct electron detector camera (Gatan Inc.) set to superresolution mode with a pixel size of 0.41 Å (a physical pixel size of 0.82 Å) and a defocus ranging from −1.3 μm to −2.0 μm. The dose rate was 20 e⁻ s⁻¹, and each movie was 1.76 s long, dose-fractioned into 40 frames, with an exposure of 1.3 e⁻ Å⁻² for each frame.

**Gene synthesis**. The gene fragments for mEGFP, mCherry, ALFA and BC2 tags, NbALFA, NbBC2, BbZIP, ZAC, hP2X3, and membrane proteins from SARS-CoV-2 used for this research were synthesized by Genewiz (Suzhou, China).

**Statistics and reproducibility**. The thermostability assays in Fig. 6c were repeated six times. Error bars represent the standard error of the mean. All FSEC experiments were repeated at least twice, and similar results were obtained.

**Reporting summary**. Further information on research design is available in the Nature Research Reporting Summary linked to this article.

## Data availability

All data and materials are available from the authors upon reasonable request. The plasmids shown in Fig. 9 (mEGFP-NbALFA, mCherry-NbALFA, pETNb-nALFA, pETNb-cALFA, pFBNb-cALFA, pBMNb-cALFA) have been deposited into Addgene (http://www.addgene.org/) (Addgene IDs: 159986, 159987, 159988, 159989, 159990 and 159991). We also deposited the BbZIP gene in pETNb-cALFA and the hP2X3 gene in pBMNb-cALFA as positive controls for FSEC-Nb (Addgene IDs 160498 and 160499). All the source data underlying the graphs and charts in the figures is available as Supplementary Data.

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

## Acknowledgements

We thank the staff scientists at the Center for Biological Imaging, Institute of Biophysics, and National Center for Protein Science Shanghai (Chinese Academy of Sciences) for technical support with cryo-EM data collection (project numbers: CBIapp201907006, CBIapp202001013 and 2019-NFPS-PT-004226) and Dr. Hideaki E. Kato (University of Tokyo) and Dr. Chia-Hsueh Lee (St. Jude Children's Research Hospital) for critical comments on the paper. This work was supported by funding provided by the Ministry of Science and Technology of China (National Key R&D Program of China: 2016YFA0502800) to M.H., by funding provided by the National Natural Science Foundation of China (32071234), and by funding provided by the Innovative Research

Team of High-level Local University in Shanghai and a key laboratory program of the Education Commission of Shanghai Municipality (ZDSYS14005).

## Author contributions

F.J. and C.S. performed experiments with assistance from M.S., M.W. and Y.W. F.J. and M.H. wrote the paper. M.H. supervised the research. All authors discussed the paper.

## Competing interests

The authors declare no competing interests.
