## [Peer Review File · Communications Biology]

Reviewers' comments:

Reviewer #1 (Remarks to the Author):

Poor biochemical behaviors continue to torment structural biologists focusing on membrane proteins. While Fluorescence-detection Size-Exclusion Chromatography (FSEC) has been useful, it is well recognized that GFP-triggered artifacts sometimes mislead the investigators. To mitigate this fundamental problem, the Hattori group developed a novel FSEC approach where GFP is cleverly attached after the expression and extraction of a target membrane protein. The authors demonstrate this novel approach works well with a bacterial membrane protein that repudiates GFP as an expression partner (ZIP), a previously crystallized eukaryotic membrane protein (P2X3), or a poorly characterized ion channel (ZAC). This robust, post-extraction GFP-tagging approach, lead to the discovery of a well-behaved ZAC orthologue (OIZAC) that can be visualized at low resolution using cryo-EM.

This paper clearly demonstrates the usefulness of their novel FSEC strategy. Though a similar approach using a His-tag binding chemical already exists (Backmark et al. 2013), the Hattori method furnishes more specific detection of the target protein, which enables the screening of weakly expressing proteins. Here are my specific comments/suggestions that may help improve this paper.

1. Except for ORF3a and Fb (Figure 5), the FSEC-Nb traces display much smaller peaks than typical traces from a conventional FSEC experiment (e.g. Supplementary Fig. 2). While this may not be an issue if FSEC traces are used only to compare the peak heights—like the authors do in this paper, it could be problematic if the peak shapes are used to gauge protein monodispersity. It is widely acknowledged that well-expressed proteins are not always monodisperse and often present shoulder peaks in FSEC experiments. Can FSEC-Nb be used to judge protein monodispersity? I suggest two things: 1) Increase the starting culture volume, which in principle should result in larger FSEC peaks with more features. Increasing the starting culture volume by 2-5 fold should not limit the usefulness of this method, as it still merely requires 10-25 ml of bacterial or 2-5 ml of mammalian cultures. 2) Use a poorly behaved protein sample to demonstrate FSEC-Nb can distinguish between monodisperse and polydisperse proteins.

2. The NbBC2 experiments (Figure 2b) are unnecessary. I suspect many other untested Nbs would also fail.

3. Page 11. The rationale of using the superfolder GFP is unclear. If this stable variant of GFP is properly folded in the periplasm, the lack of FSEC signal from the GFP-tagged BbZIP is probably attributed to steric interference of BbZIP folding or trafficking, but should be nothing to do with GFP misfolding.

4. Figure 4b and c. Scale the traces similar to "a" for better comparison. Also stronger signals for all experiments in Figure 4 (i.e. increasing the starting cultures) are necessary to highlight the advantage of FSEC-Nb. While the current Figure 4 implies that neither the conventional nor P3NTA-based FSEC can be used to screen poorly expressed ZAC proteins, the FSEC-Nb signals are also very weak. One would easily misinterpret the results and drop OIZAC, as it shows only a small peak comparable to the void peak (Figure 4a).

5. Page 21. "In our recent worst case..." I wonder FSEC-Nb would have been able to avoid such false positives. Showing the FSEC-Nb traces of TpMgtC or AtMgtC would be helpful.

Reference

Fluorescent probe for high-throughput screening of membrane protein expression.
Backmark AE, Olivier N, Snijder A, Gordon E, Dekker N, Ferguson AD. *Protein Sci.* 2013 Aug;22(8):1124-32. doi: 10.1002/pro.2297. Epub 2013 Jul 3. PMID: 23776061

Reviewer #2 (Remarks to the Author):

Fluorescence size exclusion chromatography is a useful technique to screen for membrane proteins that are tractable to structural biology. Its major disadvantages are that a) GFP must be fused to the protein and this can cause either false positives because the GFP keeps the protein in solution, or false negatives, because the GFP prevents expression b) in E coli, the GFP must be intracellular and this precludes certain proteins where the N and C termini are extracellular. A method to overcome this was previously proposed where a fluorescence probe would target a poly-histidine tag on the protein *Protein Sci* 22, 1124-1132. The problem with this method was specificity. Jin et al seek to overcome this by using a peptide tag that targets a nanobody that can be fused to a fluorophore as published in *Nature Communications* last year *Nat Commun* 10, 4403 (2019). They try two different nanobody targeting peptides but only one gives good results in the case they present. They have then tested this method on three different systems, including one that did not work with other methods. The results look convincing and they have managed to get protein to put on cryo-EM grids.

Everything looks relatively well described and convincing. I would have liked to see some discussion of whether it had been tried in other cases without success.

The major message of this paper is that the previously published fluorescently tagged antibody can help with screening of membrane proteins, overcoming the pitfalls of GFP-fusion protein. While this is a relatively small modification to previously published techniques, the demonstration that it can work may spur others in the field to try the method if the more straightforward methods fail.

Reviewer #1

“Poor biochemical behaviors continue to torment structural biologists focusing on membrane proteins. While Fluorescence-detection Size-Exclusion Chromatography (FSEC) has been useful, it is well recognized that GFP-triggered artifacts sometimes mislead the investigators. To mitigate this fundamental problem, the Hattori group developed a novel FSEC approach where GFP is cleverly attached after the expression and extraction of a target membrane protein. The authors demonstrate this novel approach works well with a bacterial membrane protein that repudiates GFP as an expression partner (ZIP), a previously crystallized eukaryotic membrane protein (P2X3), or a poorly characterized ion channel (ZAC). This robust, post-extraction GFP-tagging approach, lead to the discovery of a well-behaved ZAC orthologue (OIZAC) that can be visualized at low resolution using cryo-EM.

This paper clearly demonstrates the usefulness of their novel FSEC strategy. Though a similar approach using a His-tag binding chemical already exists (Backmark et al. 2013), the Hattori method furnishes more specific detection of the target protein, which enables the screening of weakly expressing proteins. Here are my specific comments/suggestions that may help improve this paper. ”

We appreciate the positive response from Reviewer #1. We have addressed the specific comments below.

“Except for ORF3a and Fb (Figure 5), the FSEC-Nb traces display much smaller peaks than typical traces from a conventional FSEC experiment (e.g. Supplementary Fig. 2). While this may not be an issue if FSEC traces are used only to compare the peak heights—like the authors do in this paper, it could be problematic if the peak shapes are used to gauge protein monodispersity. It is widely acknowledged that well-expressed proteins are not always monodisperse and often present shoulder peaks in FSEC experiments. Can FSEC-Nb be used to judge protein monodispersity? I suggest two things: 1) Increase the starting culture

volume, which in principle should result in larger FSEC peaks with more features. Increasing the starting culture volume by 2-5 fold should not limit the usefulness of this method, as it still merely requires 10-25 ml of bacterial or 2-5 ml of mammalian cultures. 2) Use a poorly behaved protein sample to demonstrate FSEC-Nb can distinguish between monodisperse and polydisperse proteins.”

We appreciate the comments from Reviewer #1. Yes. FSEC-Nb can be used to judge the protein monodispersity by comparing peak shapes, as in the case with the conventional GFP fusion-based FSEC. According to the comments, we performed both experiments, as Reviewer #1 suggested.

“1) Increase the starting culture volume, which in principle should result in larger FSEC peaks with more features. Increasing the starting culture volume by 2-5 fold should not limit the usefulness of this method, as it still merely requires 10-25 ml of bacterial or 2-5 ml of mammalian cultures.”

We performed FSEC-Nb experiments on ALFA-tagged BbZIPs in *E. coli* cells by increasing the starting culture volume (5 ml, 10 ml, and 20 ml). As expected, the FSEC-Nb plots showed higher peaks with more shape features as the culture volume increased (**Fig. 2d**) (Page 10, Lines 6-8). We also performed FSEC-Nb experiments of ALFA-tagged hP2X3 in HEK293 cells by increasing the starting culture volume (1 ml, 2 ml, and 4 ml), which resulted in higher peaks of the target as the culture volume increased (**Fig. 3c**) (Page 11, Lines 5-8). Intriguingly, as the peak became higher, the elution position of the peak shifted to an elution position corresponding to a lower molecular weight (**Fig. 3c**). This outcome was probably obtained because as the culture volume increased, the fractions of the P2X3 trimer, which does not have three full mEGFP-tagged NbALFA bound per P2X3 trimer, increased, yielding a shift of the elution position to a position corresponding to a lower molecular weight (Page 11, Lines 8-13). We have added new figures and corresponding descriptions to the revised manuscript.

“2) Use a poorly behaved protein sample to demonstrate FSEC-Nb can distinguish between monodisperse and polydisperse proteins.”

According to this comment, we performed FSEC-Nb experiments to screen ALFA-tagged ZAC homologs by increasing the starting culture volume (1 ml, 2 ml, and 4 ml), which resulted in not only higher peaks but also clearer peak shapes from ZAC homologues (**Fig. 4**). With 2 ml and 4 ml of the starting culture volume, OIZAC exhibited the high monodispersity of the main peak in FSEC-Nb (**Fig. 4b, c**). In contrast, the FSEC plots of OnZAC showed a much broader peak with the left shoulder in FSEC-Nb (**Fig. 4b, c**) (From Page 13, Line 15 to Page 14, Line 3). Consistently, we could not purify OnZAC as it aggregated during purification. Overall, FSEC-Nb can be used to distinguish protein samples with good behaviour from poorly behaved samples based on the monodispersity of peak shapes in FSEC-Nb (Page 14, Lines 3-7). We have added a new figure and the corresponding descriptions in the revised manuscript (**Fig. 4a, b, c**) (From Page 13, Line 14 to Page 14, Line 7).

“2. The NbBC2 experiments (Figure 2b) are unnecessary. I suspect many other untested Nbs would also fail.”

We appreciate and fully respect the comment and agree that whether the NbBC2 experiments are included would not affect our main conclusion. However, since Reviewer #2 would like us to show our unsuccessful trials (see below), we would like to retain the NbBC2 experiments, which may be beneficial to some of the readers who plan to develop an FSEC-Nb derivative method in the future.

“3. Page 11. The rationale of using the superfolder GFP is unclear. If this stable variant of GFP is properly folded in the periplasm, the lack of FSEC signal from the GFP-tagged BbZIP is probably attributed to steric interference of BbZIP folding or

trafficking, but should be nothing to do with GFP misfolding.”

We appreciate the comment. In response to this comment, we have removed the superfolder GFP experiments and the descriptions from our revised manuscript.

“4. Figure 4b and c. Scale the traces similar to “a” for better comparison. Also stronger signals for all experiments in Figure 4 (i.e. increasing the starting cultures) are necessary to highlight the advantage of FSEC-Nb. While the current Figure 4 implies that neither the conventional nor P3NTA-based FSEC can be used to screen poorly expressed ZAC proteins, the FSEC-Nb signals are also very weak. One would easily misinterpret the results and drop OIZAC, as it shows only a small peak comparable to the void peak (Figure 4a).”

We appreciate this comment. To respond to this comment, we scaled the traces of P3NTA-based FSEC similar to those of FSEC-Nb for better comparison (**Fig. 4d**). The experiments of superfolder GFP-tagged OIZAC have been removed in the revised manuscript, as we mentioned above. Furthermore, according to this comment, we tested the FSEC-Nb of ZAC proteins by increasing the starting culture volume as described above to show higher peaks with more shape features, as we mentioned above (**Fig. 4a, b, c**).

“5. Page 21. “In our recent worst case...” I wonder FSEC-Nb would have been able to avoid such false positives. Showing the FSEC-Nb traces of TpMgtC or AtMgtC would be helpful.”

We appreciate the comment. We performed FSEC-Nb experiments on TpMgtC and AtMgtC, yielding poor profiles. We have added the results and corresponding descriptions to the revised manuscript (**Supplementary Fig. 2c, d**) (Page 21, Lines 11-13).

Reviewer #2

“Fluorescence size exclusion chromatography is a useful technique to screen for membrane proteins that are tractable to structural biology. Its major disadvantages are that a) GFP must be fused to the protein and this can cause either false positives because the GFP keeps the protein in solution, or false negatives, because the GFP prevents expression b) in E coli, the GFP must be intracellular and this precludes certain proteins where the N and C termini are extracellular. A method to overcome this was previously proposed where a fluorescence probe would target a poly-histidine tag on the protein Protein Sci 22, 1124-1132. The problem with this method was specificity. Jin et al seek to overcome this by using a peptide tag that targets a nanobody that can be fused to a fluorophore as published in Nature Communications last year Nat Commun 10, 4403 (2019). They try two different nanobody targeting peptides but only one gives good results in the case they present. They have then tested this method on three different systems, including one that did not work with other methods. The results look convincing and they have managed to get protein to put on cryo-EM grids. ”

We appreciate the positive response from Reviewer #2. We have addressed the specific comments below.

“Everything looks relatively well described and convincing. I would have liked to see some discussion of whether it had been tried in other cases without success. ”

We appreciate the positive comment. We tested ALFA and BC2 tags for FSEC-Nb but did not test other peptide tags and their nanobodies for FSEC-Nb. Therefore, we do not know whether other peptide tags can be applied successfully for FSEC-Nb (Page 20, Lines 3-5). Regarding the application of FSEC-Nb with an ALFA tag to other proteins in addition to ZAC and SARS-CoV-2 proteins, we tested the expression of MgtC by FSEC-Nb (**Supplementary Fig. 2c, d**). Whereas the conventional

GFP-fusion FSEC identified two MgtE proteins with high monodispersity and expression levels (**Supplementary Fig. 2a, b**), both proteins aggregated and precipitated after the removal of the GFP tag. In contrast, we did not detect the expression of these two proteins at the corresponding positions by FSEC-Nb (**Supplementary Fig. 2c, d**). We have added new figures and corresponding descriptions to the revised manuscript (Page 21, Lines 11-13).

“The major message of this paper is that the previously published fluorescently tagged antibody can help with screening of membrane proteins, overcoming the pitfalls of GFP-fusion protein. While this is a relatively small modification to previously published techniques, the demonstration that it can work may spur others in the field to try the method if the more straightforward methods fail.”

We appreciate another positive response from Reviewer #2. We hope this method will be useful to the community.

REVIEWERS' COMMENTS:

Reviewer #1 (Remarks to the Author):

The authors appropriately addressed all the concerns raised by both reviewers. I have no further comments.

Reviewer #2 (Remarks to the Author):

The concerns of the referees look to have been addressed fully.